# Tailoring π-conjugation and vibrational modes to steer on-surface synthesis of pentalene-bridged ladder polymers

Bruno de la Torre [1,2,6], Adam Matěj [1,2,6], Ana Sánchez-Grande[3,6], Borja Cirera[3], Benjamin Mallada [1,2], Eider Rodríguez-Sánchez [3], José Santos [3,4], Jesús I. Mendieta-Moreno[2], Shayan Edalatmanesh [1,2], Koen Lauwaet[3], Michal Otyepka [1], Miroslav Medveď [1], Álvaro Buendía[5], Rodolfo Miranda[3,5], Nazario Martín [3,4✉], Pavel Jelínek [1,2✉] & David Écija[3✉]

The development of synthetic strategies to engineer π-conjugated polymers is of paramount importance in modern chemistry and materials science. Here we introduce a synthetic protocol based on the search for specific vibrational modes through an appropriate tailoring of the π-conjugation of the precursors, in order to increase the attempt frequency of a chemical reaction. First, we design a 1D π-conjugated polymer on Au(111), which is based on bisanthene monomers linked by cumulene bridges that tune specific vibrational modes. In a second step, upon further annealing, such vibrational modes steer the twofold cyclization reaction between adjacent bisanthene moieties, which gives rise to a long pentalene-bridged conjugated ladder polymer featuring a low bandgap. In addition, high resolution atomic force microscopy allows us to identify by atomistic insights the resonance form of the polymer, thus confirming the validity of the Glidewell and Lloyd´s rules for aromaticity. This on-surface synthetic strategy may stimulate exploiting previously precluded reactions towards π-conjugated polymers with specific structures and properties.

[1] Regional Centre of Advanced Technologies and Materials, Palacký University, Šlechtitelů 27, 783 71 Olomouc, Czech Republic. [2] Institute of Physics, The Czech Academy of Sciences, Cukrovarnická 10, 162 00, Prague 6, Czech Republic. [3] IMDEA Nanociencia, C/ Faraday 9, Ciudad Universitaria de Cantoblanco, 28049 Madrid, Spain. [4] Departamento de Química Orgánica, Facultad de Ciencias Químicas, Universidad Complutense, 28040 Madrid, Spain. [5] Departamento de Física de la Materia Condensada, Universidad Autónoma de Madrid, Cantoblanco, Madrid, Spain. [6] These authors contributed equally: Bruno de la Torre, Adam Matěj, Ana Sánchez-Grande. ✉email: nazmar@ucm.es; jelinekp@fzu.cz; david.ecija@imdea.org

The design and synthesis of π-conjugated polymers is a very active area of research with great potential for applications in organic field-effect transistors (OFETs), photovoltaics (OPVs), and light emitting diodes (OLEDs)[1–3].

However, the advance in the synthesis of π-conjugated polymers is hampered by concomitant limitations of solubility during the chemical synthesis[2,3]. This drawback is even enhanced in the design of π-conjugated ladder polymers, a singular type of polymers in which all the backbone is π-conjugated and fused[4]. These conjugated ladder polymers are of great appeal for materials science and optoelectronics due to their exceptional stability and optimum electron delocalization thanks to the restriction of free torsional motion in between monomers[4].

In the field of π-conjugated polymers, the topology of the π-electron network is crucial since it determines the ground state electronic structure of such materials. Polymers incorporating non-benzenoid polycyclic hydrocarbons are of increasing interest due their specific electronic properties[5]. In non-benzenoid systems, molecular orbital levels and π-electron density distribution are uneven compared to benzenoid systems, thereby polarizing the ground state and leading to unique behavior in excited states[5]. π-Extended non-benzenoid polycyclic hydrocarbons, such as pentalene, indacene or indenofluorene, have recently propelled rich insights into the electronic properties of antiaromaticity and/or open-shell character[6–12]. On the one hand, antiaromatic compounds show better conducting properties than their aromatic counterparts[13]. On the other hand, non-benzenoid species are prone to express open-shell character in solution[6,7] and on surfaces[9–12].

Taking into account the aforementioned unique properties, it is of timely relevance to engineer conjugated ladder polymers incorporating non-benzenoid components, targeting to design chemically robust and low bandgap polymers. Such synthesis in wet chemistry must contend with structural defects and low solubility that prevent complete control over the synthesis and structural characterization at the atomic scale. To overcome these synthetic challenges, on-surface synthesis has emerged as a disruptive paradigm to develop chemical reactions on surfaces, while simultaneously monitoring and elucidating the precursors, intermediates and reaction products by means of scanning probe microscopy[14–18]. Despite the recent progress in on-surface synthesis, there are still very limited strategies to synthesize complex π-conjugated polymers with specific properties[14–27].

The electronic and chemical properties of π-conjugated polymers are driven by the inherent topology of their π-electrons. Here, we present a chemical strategy to allow unconventional synthetic pathways by exploiting the relation between π-conjugation and specific vibrational modes, being able to steer a desired chemical transformation. Our strategy is based on two sequential steps, i.e., a polymerization expressing specifically tailored π-topology, and a subsequent complex chemical ladderization involving a twofold cyclization by taking advantage of the unique vibrational capabilities of the inherent π-topology. In the first step, the on-surface formation of a one-dimensional polymeric precursor featuring cumulene bridges takes place, allowing bending vibrational modes of the polymer with a desired directionality. In the second and final steps, subsequent thermal annealing drives intrapolymeric cyclization reactions, triggered by such specific vibrational modes, giving rise to ladderization of the polymer and the expression within the polymeric main-chain of non-benzenoid fused pentalene bridges.

## Results

### Synthesis of pentalene-bridged bisanthene. Following this chemical strategy, we report the synthesis and detailed atomistic characterization by a comprehensive scanning tunneling microscopy (STM) and non-contact atomic force microscopy (nc-AFM) supported by density functional theory (DFT) study of the on-surface reactions of two distinct molecular precursors, **4BrBiA** [10,10′-bis(dibromomethylene)-10H,10′H-9,9′-bianthracenylidene] and **4BrAn** [9,10-bis(dibromomethylene)-9,10-dihydroanthracene]. The precursors are endowed with $=CBr_2$ functionalities to allow homocoupling[23,27]. The deposition of **4BrAn** and **4BrBiA** on Au(111) and subsequent annealing at 500 K results in the formation of anthracene (**1**, Fig. 1a) and bisanthene (**2**, Fig. 1b) polymers, respectively[23,27]. Both polymers exhibit repeating moieties linked by linear bridges, but with distinct π-conjugation. While the anthracene polymer adopts aromatic ethynylene π-conjugation character[23], the bisanthene polymer results in the promotion of the quinoid cumulene-like character[27].

Further annealing at 650 K leads to distinct behavior of the formed polymers despite their chemical similarity. Our theoretical calculations (see later) reveal that, as a result of this different π-topology, the cumulene-like bridge (with a weakened triple bond) in the bisanthene polymer (**2**) allows the amplification of two bending vibrational modes of the bridging unit (see Fig. 2a), which promotes a distinct reaction mechanism with respect to the anthracene polymer (**1**) (see Fig. 1a and Supplementary Fig. 1). In the case of the cumulene-bridged bisanthene polymer (**2**) (see Fig. 1c) these vibrational modes substantially overlap with the intrapolymeric reaction coordinate. Their thermal excitation upon annealing at 650 K promotes the formation of fused pentalene bridges, which gives rise to a long and defect-free π-conjugated ladder polymer (**3**), as shown in Fig. 1d. High-resolution nc-AFM images acquired with CO-tip reveal that the fused bridging units are formed by pentalene moieties. Notably, this intrapolymeric ladderization reaction is highly selective and only minority concomitant defects are detected for submonolayer coverage. On the one hand, for the case of the anthracene polymer (**1**), annealing above 700 K results in distinct side reactions giving rise to irregular polymers with variety of defects (see Supplementary Fig. 1). Thus, the annealing of cumulene-linked bisanthene polymers allows chemical reactions that are precluded for the ethynylene-bridged anthracene wires.

### Reaction mechanism of the on-surface synthesis of pentalene-bridged bisanthene. To get a deeper insight into these differences and the underlying reaction mechanism, we carried out quantum mechanical/molecular mechanics (QM/MM) simulations[28,29] with the aim of analyzing the optimal reaction pathways of the bridging units of both anthracene and bisanthene polymers towards the formation of pentalene links, for more details of computational model see section "Methods". In general, the feasibility of a chemical reaction is determined by two factors: (i) the Boltzmann exponential factor given by the activation energy barrier, and (ii) the attempt frequency $\nu$ expressing how often the system evolves towards the saddle point of the reaction pathway[30–33]. In the following, we will show that the π-conjugation form adopted by the polymer may also strongly influence the optimal reaction pathway by tuning both the activation energy and steering vibrational modes.

To find the most feasible reaction pathway towards the formation of the pentalene-bridged polymer (**3**), we attempted different reaction coordinates mimicking the first step of the reaction. Supplementary Fig. 2 displays two of the most relevant reaction coordinates. Namely, we have considered either a direct cyclization between the bridges and the rims of the monomers (see Fig. 2c) or a dissociation of the C–H bond at the zigzag edge of the bisanthene monomer mediated by an Au adatom (see

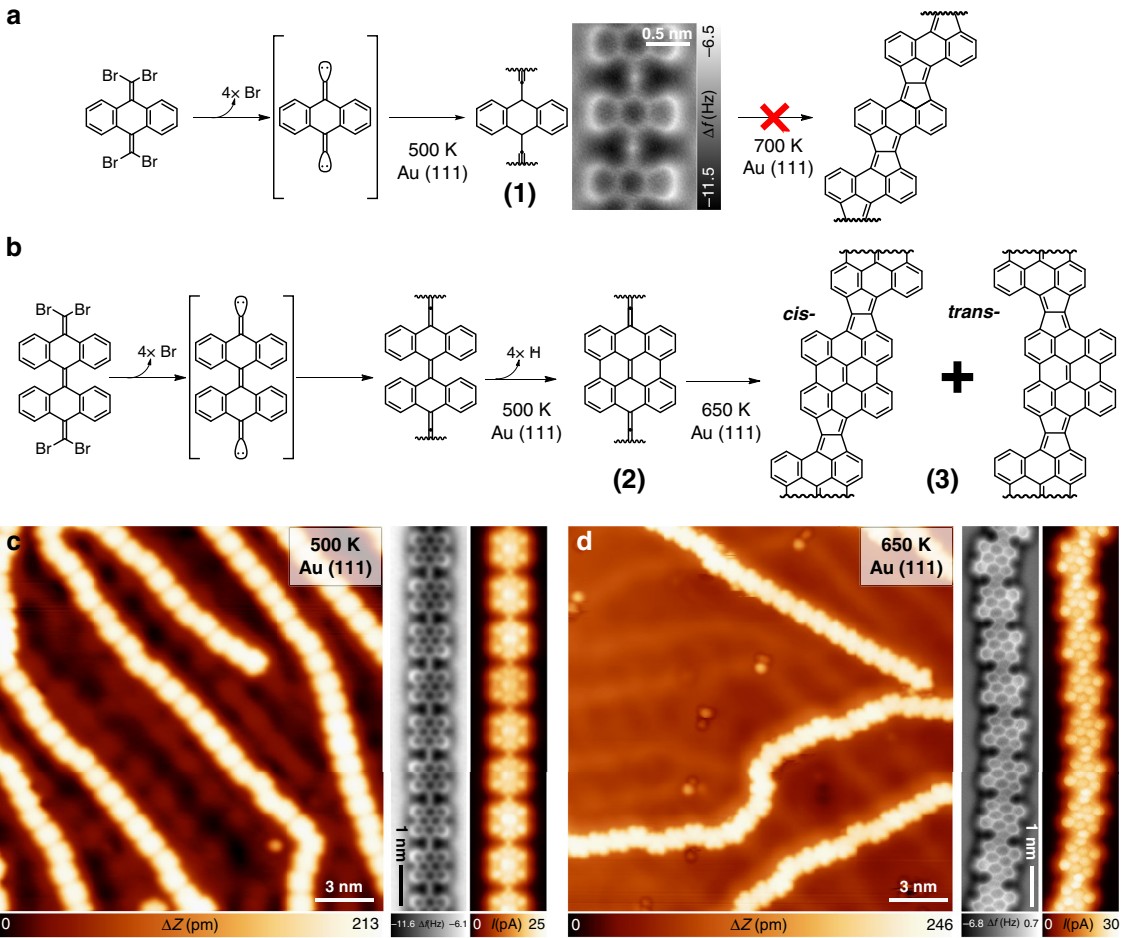

**Fig. 1 On-surface synthesis of pentalene-bridged bisanthene polymers. a** Scheme of the reaction sequence of **4BrAn** precursor after being deposited on Au(111), annealed to 500 K to obtain polymer **1**, and annealed up to 700 K to only produce irregular defective polymers (see Supplementary Fig. 1). **b** Scheme of the reaction sequence of **4BrBiA** precursor after being deposited on Au(111), annealed to 500 K to obtain polymer **2**, and annealed to 650 K to result in polymer **3**. **c** STM topographic overview (left panel; 0.03 V, 0.1 nA) and detailed nc-AFM/STM constant height images (right panels) of cumulene-linked bisanthene polymers (**2**). **d** STM topographic overview (left panel; 0.05 V, 0.1 nA) and detailed nc-AFM/STM constant height images (right panels) of pentalene-bridged bisanthene polymers (**3**).

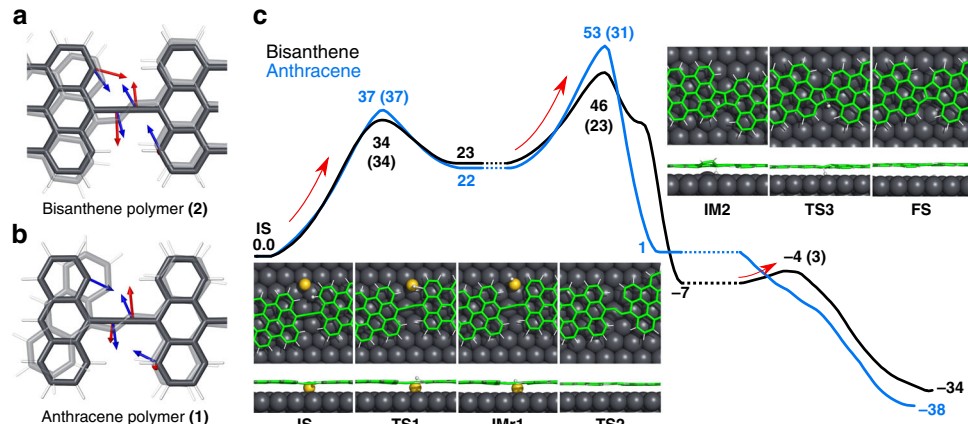

**Fig. 2 Calculated vibrational modes and reaction pathways of polymers (1) and (2) on Au(111). a, b** Comparison of calculated vibrational mode (red arrows) with reaction coordinates (blue arrows) for **a**, cumulene-like bridged bisanthene polymer (**2**) and for **b**, ethynylene-linked anthracene polymer (**1**). The atomic arrangement of the transition state (**TS2**) is showed in transparent mode. Reaction coordinates are defined as the displacement from the initial (**IMr1**) to the transition state (**TS2**) for the cyclization reaction pathway from **IMr1** to **IM2**. **c** Free energy barriers (in kcal mol$^{-1}$) for the optimal reaction pathway calculated for anthracene (**1**, blue) and bisanthene (**2**, black) polymers at 650 K. Values of the activation energies are presented in brackets. Inset figures show top and side views of the initial state (**IS**), transition states (**TS**), intermediates (**IM**) and final states (**FS**) of bisanthene polymer (**2**) optimal reaction pathway towards pentalene-bridged bisanthene polymer (**3**).

Supplementary Fig. 2d). The QM/MM calculations at 650 K show that the activation free energy for hydrogen abstraction (34 and 37 kcal mol$^{-1}$) is much lower than that of the direct cyclization (82 and 110 kcal mol$^{-1}$) for both bisanthene (**2**) and anthracene (**1**) polymers, respectively (see Supplementary Figs. 3 and 4). Thus, we assume that the first step of the reaction pathway is a hydrogen abstraction in both cases.

In the second step of the reaction, the cyclization process for both polymers (**1** and **2**) have slightly lower activation barrier than a second dehydrogenation by a few kcal mol$^{-1}$ (see Supplementary Figs. 3 and 4). The calculated activation free energy of cyclization for the bisanthene polymer (**2**) is significantly smaller (23 kcal mol$^{-1}$) than that for the anthracene polymer (**1**, 31 kcal mol$^{-1}$). As the activation barriers of the cyclization process in the second reaction step are lower than those corresponding to the second hydrogen abstraction, the dehydrogenation-cyclization-dehydrogenation scenario (shown in Fig. 2 and also denoted by yellow lines in Supplementary Figs. 3 and 4) seems to be the optimal reaction pathway among the explored reaction pathways for both polymers. Nevertheless, experimentally we only observe the formation of the long and defect-free pentalene-bridged polymers from the bisanthene (**2**), but not from the anthracene (**1**) polymers. Thus, the activation energies cannot explain by themselves the experimental evidence.

However, the reaction rate also depends on the $\nu$ prefactor, which is determined by available steering vibrational eigenmodes strongly overlapping with the reaction coordinates. Figure 2a, b shows the atomic arrangement in the transition state (shown in translucent color) of the cyclization reaction, which involves a bending distortion of the cumulene bridge. Thus, we carried out the dynamical matrix calculations of all vibrational modes of freestanding bisanthene/anthracene polymers, from which we selected only vibrational modes with dominant contribution of bending mode on the bridging linker adjacent to the periacene/ acene unit with unsaturated carbon atom, see Supplementary Fig. 5. According to our analysis of the vibrational modes of bisanthene polymer (**2**), we can identify two bending vibrational modes that are coupled with the in-phase displacement of the unsaturated carbon atom of the bisanthene moiety (see red arrows in Fig. 2a). Such bending vibrational modes match very well with the cyclization reaction coordinate depicted by blue arrows in Fig. 2a (also see vibrational modes of bisanthene labeled 182 and 185 shown in Supplementary Fig. 5).

On the contrary, in the case of the anthracene polymer (**1**), all the vibrational modes consisting of the bending mode lack coupling with the in-phase movement of the unsaturated carbon atom (see Fig. 2b). Therefore, in the case of the cumulene-like bridged bisanthene polymer (**2**) the strong overlap of the bending vibrational modes with the cyclization reaction coordinates together with slightly higher vibrational frequencies provide larger attempt frequency $\nu$ than in the case of ethynylene-linked anthracene polymer (**1**). Indeed, the bending modes of ethynylene-bridge of the anthracene polymer (**1**) are completely lack of the in-phase motion of the unsaturated carbon atoms, see Supplementary Fig. 5, which plays the important role in the formation of the pentalene bridge. As a result, the overlap between the cyclization reaction coordinate and the internal vibrational modes of the polymer is significantly reduced with respect to the bisanthene case. Consequently, this causes substantial drop of the corresponding attempt frequency $\nu$. Furthermore, the anthracene polymer (**1**) contains the ethynylene bridges, thus featuring a triple bond, which has distinct bonding character than the cumulene-like linkers of bisanthene polymer (**2**)[27]. Therefore, the frequencies of the steering bending modes of the ethynylene linkers in anthracene polymers (**1**) are lower by 50–100 cm$^{-1}$ (i.e., a drop of ~10%) than those of cumulene-like

bridges in bisanthene polymers (**2**), as shown in Supplementary Fig. 5, leading to a further decrease of the attempt frequency.

In summary, the distinct $\pi$-conjugation of the bisanthene polymer (**2**) exhibiting the cumulene-like bridge is prone to the bending vibrational modes that steer the cyclization reaction. These bending vibrational modes coupled with the appropriated in-phase motion of bisanthene moiety enhance not only the attempt frequency $\nu$, but they also impose a lower energy cost of the bond distortion contributing to the activation energy than for the anthracene polymer (**1**). Altogether, these two factors facilitate the formation of the defect-free pentalene-bridged bisanthene polymers (**3**) as observed experimentally.

At this point, it is worth to point out that the hydrogenation abstraction is driven by the C–H stretching mode (~3100 cm$^{-1}$), which is approximately one order of magnitude larger than the frequencies of the C–C bending modes (~600 cm$^{-1}$), see Supplemental Fig. 5. Thus, the $\nu$ prefactor and, consequently, the total rate of the dehydrogenation processes are enhanced. In the case of anthracene polymer (**1**), the energy barrier (36 kcal mol$^{-1}$) for the second dissociation is only slightly higher than that of the cyclization (31 kcal mol$^{-1}$) (see Supplementary Fig. 4). Consequently, considering that the steering C–H stretching mode is approximately one order of magnitude higher than the C–C bending modes, the reaction rates for the second dissociation or the cyclization at 700 K become similar. This opens a second reaction channel that includes a subsequent double dehydrogenation and cyclization reaction trajectory (denoted as a blue line in Supplementary Fig. 4). Notably, for this reaction trajectory of the anthracene polymer (**1**), the formation of the pentalene bridge occurs through quite distinct transition state (compare TS4 in Supplementary Fig. 4 with TS6 in Supplementary Fig. 3 for the bisanthene polymer) featuring very large activation energy (>100 kcal mol$^{-1}$), not shown in Supplementary Fig. 4. The QM/MM simulations indicate that unsaturated carbon atoms in the anthracene unit connect to the nearest carbon atoms on the ethynylene-bridge, which leads to a different atomic arrangement than in the bisanthene polymer. Such a process can be understood by the absence of the in-phase motion of the bending vibrational mode with the distant unsaturated carbon atoms of the anthracene unit as discussed previously. This result clearly demonstrates that the cyclization reaction coordinates forming the pentalene bridge from the ethynylene-bridged anthracene polymer (see Supplementary Fig. 2d) are no longer the optimal ones and other more favorable reaction mechanisms are presented. Thus, a formation of other defective structures is preferred, which is experimentally confirmed and illustrated in Supplementary Fig. 1. Such findings mean that other reaction pathways, including the double H-dissociation and cyclization, with lower activation energies and/or suitable active vibrational modes matching corresponding reaction coordinates, take place avoiding the formation of the pentalene-bridged anthracene polymers.

The discussion above points out the importance of the internal modes of the reactant on the chemical rate of a given reaction. From this perspective, it is worth to make a connection to well-established transition state theory[34], which is widely adopted to describe the chemical rates of diverse reaction mechanisms. Strictly speaking, in the transition state theory the rate of a reaction is determined by dynamics of the system at the transition state (i.e., dividing surface, which separates the phase space into a reactant and product region, placed at the lowest saddle point of potential energy surface). Importantly, one of the basic assumptions of the transition state theory is the existence of a quasi-equilibrium in the reactant region (i.e., between the reactants and the activated complex formed at the transition state). This assumption means that the reactants have sufficient kinetic

energy, which allows to reach the transition state (i.e., to form the activation complex) at certain frequency to establish the quasi-equilibrium. In the case of bimolecular reactions in gas phase, the required kinetic energy is often available through collision events of reactants. Therefore, such equilibrium can be reached depending on collision rates driven by temperature of system or kinetic energy of a molecular-beam. However, in the case of unimolecular reactions caused by an internal chemical transformation of the reactant, such as the one discussed above, the kinetic energy is driven mostly by internal vibrational modes of the reactant. Moreover, only vibrational modes with certain overlap with the reaction coordinates may contribute to the kinetic energy required to reach the transition state. Thus, it seems that not only the transition state, but also the internal vibrational modes of the reactant may play important role in such reaction mechanisms. We feel that our combined experimental and theoretical work provides a strong indication that, indeed, the vibrational modes of the initial state may play significant role on the rate. It is worth to point out that our rationalization roots back to early works from Polanyi introducing a concept of "early" and "late" reaction barriers[33], or, more recently, to control of reaction dynamics via excitations of reactant vibrational modes[31], and to modified versions of the transition state theory[35], including mode-specific reaction dynamics[32]. However, these recent theories have been tested to bimolecular reactions including reactants, much simpler than the polymers presented here.

**Structural and the electronic properties of pentalene-bridged bisanthene polymers**. Next, we focus on the structural and the electronic properties of pentalene-bridged bisanthene polymers (**3**), including the dominant resonance form. In general, the bisanthene-pentalene-bisanthene connections can feature *trans* or *cis* (Fig. 1b) configurations. A statistical analysis of the abundance of the covalent motifs reveals that both configurations are equally favorable. However, segments having repeating *trans* or *cis* motifs were found with a maximum length of twelve units.

Clar's π-sextet rule, an extension of Huckel's rule for polycyclic aromatic hydrocarbons (PAHs), states that the Kekulé resonance structure with the largest number of disjoint aromatic π-sextets, i.e., benzene-like moieties, is the most important resonance structure for the characterization of the PAHs properties. Since our pentalene-bridged polymers present non-benzenoid units, the rule to be used here is an extension of Clar's π-sextet rule, the one proposed by Glidewell and Lloyd[36,37]. This rule states that the total population of π-electrons in conjugated polycyclic systems tends to form the smallest $4n + 2$ groups and to avoid the formation of the smallest 4n groups. The pentalene bridge can show two types of resonance forms, i.e., the inner bond can be single or double. From this assumption, at least two distinct resonance forms can be plotted as illustrated in Fig. 3a. By applying the Glidewell and Lloyd´s rule, it is evident that the conjugated structure associated with an inner single bond (black structure in Fig. 3a) should be the most stable, since the other option (gray structure in Fig. 3a) would imply the formation of four groups with 4π-electrons in the bisanthene moiety, which must be avoided according to the rule. Interestingly, by following the observed π-conjugation, the system stabilizes four Clar's sextets in the bisanthene (depicted in blue in Fig. 3a), the maximum number.

To assess the expression of the Glidewell and Lloyd's rule on surfaces, we take advantage of the capabilities of nc-AFM with functionalized CO-tip[38,39] to resolve the C–C bond order[40]. High-resolution nc-AFM images of the pentalene-bridged polymers (see Fig. 3b) clearly show distinct bond lengths within the inner part of the polymer, whose statistically average value is plotted in the central panel of Fig. 3b (see Supplementary Fig. 6 for more details). The variation of bond distances is qualitatively confirmed by DFT calculations (right panel of Fig. 3b). The results reveal that the variation in the bond length of the polymer bonds matches the π-resonance form predicted by the Glidewell and Lloyd's rule, confirming its validity.

Next, the degree of aromaticity and antiaromaticity of the bisanthene polymer is characterized by performing nucleus-independent chemical shift calculations (NICS). Positive values of NICS indicate antiaromaticity, close-to-zero values reveals non-aromaticity, and negative figures correspond to aromaticity. Figure 3c shows that the five-membered rings are highly antiaromatic, whereas the outer six-membered rings of the bisanthene moieties are aromatic, thus corroborating the resonance form discussed above.

In order to elucidate the electronic structure of the π-conjugated polymers, we first calculated the electronic structure of freestanding polymer using B3LYP-DFT, which revealed the presence of dispersive valence (VB) and conduction (CB) bands separated by a bandgap of ~1 eV (Fig. 3d). Spatially resolved d$I$/d$V$ scanning tunneling spectra were recorded over the molecular wires and the clean Au(111) surface. As illustrated in Fig. 3e, one frontier resonance is distinguished at −0.6 eV, assigned to the valence band (VB) edge and a broad peak with an elbow at 0.5 eV, interpreted as the conduction band (CB) edge, displaying a maximum at 0.9 eV. This results in a low bandgap of ~1.1 eV. We should note that the bandgap value obtained from STS measurements is typically reduced by an additional electron screening imposed by the proximity of a metallic surface with respect to the intrinsic bandgap of the gas phase polymer[41,42]. Within the bandgap the downshifted surface state convoluted with tip states is observed. The d$I$/d$V$ map at the VB edge shows maxima over lateral termini of the bisanthene moiety, without charge density over the pentalene bridge (see Fig. 3f). The d$I$/d$V$ map of CB edge exhibits states over the armchair region of the bisanthene units and on the void spaces adjacent to the pentalene bridges (see Fig. 3f). Despite the fact that DFT simulations cannot predict correctly the magnitude of the intrinsic bandgap of the polymer[43], they describe the character of frontier orbitals of the VB and CB edges of the bisanthene polymer very well. Indeed, the agreement between experimental and simulated d$I$/d$V$ maps is excellent, validating the character of the frontier orbitals predicted by DFT. Thus, the fused pentalene-bridged bisanthene polymers are low bandgap polymers, featuring antiaromatic bridging units, heralding potential for near-infrared activity, high conductivity, and ambipolar charge transport[44–46].

## Discussion

Our results introduce the importance of tailoring π-conjugation and vibrational modes on surfaces to promote otherwise precluded chemical reaction pathways. Following this strategy, we reveal that a cumulene-like bridged bisanthene polymer is prone to undergo ladderization into a fused pentalene-linked bisanthene polymer through a twofold cyclization process, thanks to specific steering vibrational modes. The absence of such modes in an ethynylene-linked anthracene polymer featuring similar coordination environment, but very distinct π-topology of the linker, blocks the ladderization reaction and reveals the hidden and relevant influence of the resonance form of the bridge in the chemical reaction. Moreover, the presented study indicates that not only the transition state, but also the internal vibrational modes of the reactant may play important role in reaction

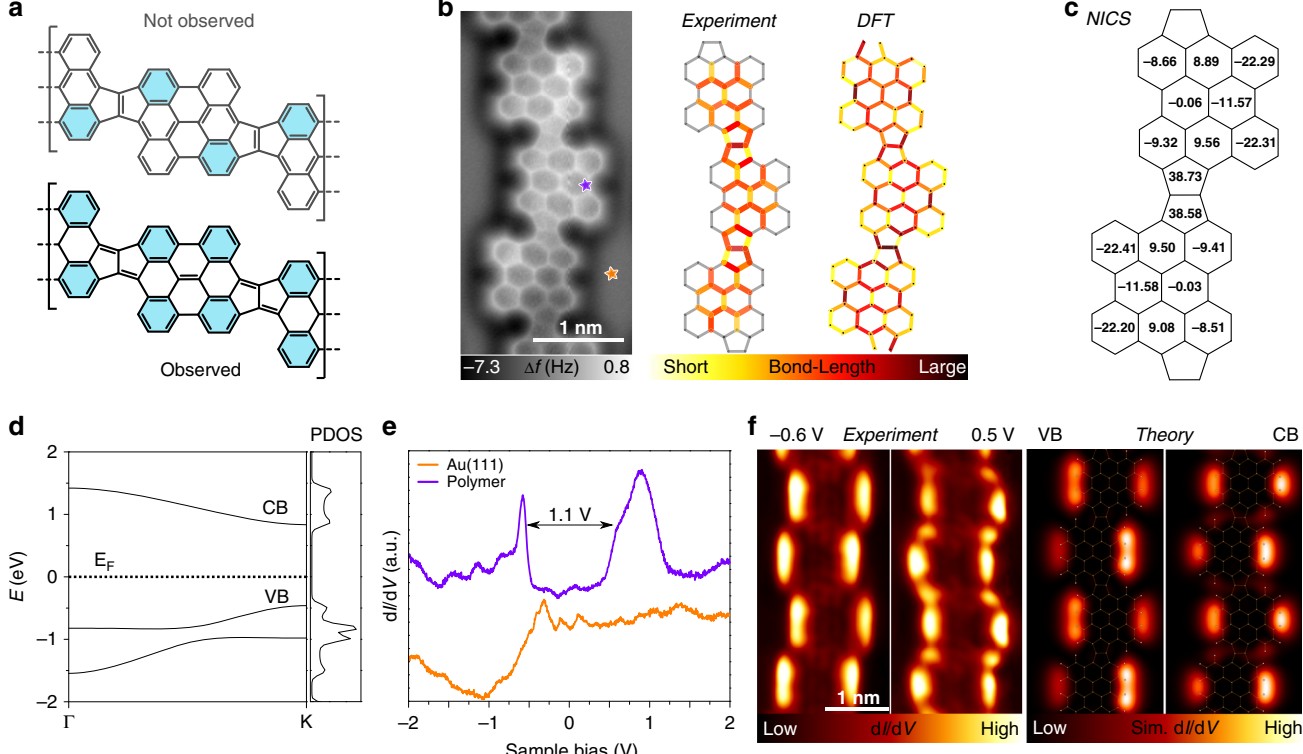

**Fig. 3 Resonance and electronic structure of the pentalene-bridged bisanthene polymer (3). a** Not observed and observed resonance forms of polymer **3**, featuring *cis* configuration. **b** High-resolution nc-AFM image of a section of the pentalene-bridged bisanthene polymer (left panel) and sketch illustrating the average bond lengths statistically measured from experimental nc-AFM images (central panel) and DFT calculation (right panel) of the bond lengths of a *trans*- section of the polymer. **c** NICS calculations of a pentalene-linked bisanthene dimer at distance of 1 Å above the center of each ring. **d** Calculated electronic structure of a freestanding cumulene-like bridged bisanthene polymer (**3**). **e** d*I*/d*V* scanning tunneling spectra acquired at selected site on polymer **3** and on Au(111) as indicated by stars in **b**. **f** Experimental and simulated d*I*/d*V* maps of the valence (VB) and conduction band (CB) of polymer **3**.

mechanisms. As this, we hope that our findings will stimulate new direction of both experimental and theoretical activities towards better understanding of the chemical reaction rates, especially those on-surface reactions driven by internal vibrational modes.

We envision our studies will open avenues to engineer highly demanded conjugated nanomaterials, while showing strategies to incorporate non-benzenoid moieties in polymeric science in order to steer relevant electronic phenomena of interest for molecular optoelectronics and organic solar cells. Notably, our results highlight the versatility of on-surface synthesis for designing non-benzenoid based nanomaterials featuring low bandgaps, which are of utmost interest because of their near-infrared activity and ambipolar charge transport character[44–46].

## Methods

**Sample preparation and experimental details.** Experiments were performed in two independent custom designed ultra-high vacuum systems that host a low-temperature Omicron and a Createc scanning tunneling microscope, respectively, where the base pressure was below $5 \times 10^{-10}$ mbar. STM images were acquired with electrochemically etched tungsten tips or cut and sharpened by focus ion beam (FIB) Pt/Ir tips, applying a bias ($V_b$) to the sample at a temperature of ~4 K. Precursor molecules were synthesized in our group following a procedure described in refs. [19–21]. The Au(111) substrate was prepared by standard cycles of Ar$^+$ sputtering (800 eV) and subsequent annealing at 723 K for 10 min. Molecular precursor were deposited by organic molecular-beam epitaxy (OMBE) from a quartz crucible maintained at 373 K (**4BrAn**) or 443 K (**4BrBiAn**) onto a clean Au (111) held at room temperature. Whenever necessary samples were annealed to the desired temperature for 20 min and subsequently transferred to the STM stage, which was maintained at 4.2 K. For the spectroscopic measurements, specific site d*I*/d*V* and d*I*/d*V* maps were acquired with conventional lock-in technique with a modulation of 5 and 10 mV, respectively. In nc-AFM imaging, a Pt/Ir tip mounted on a qplus sensor (resonance frequency ≈30 kHz; stiffness ≈ 1800 N m$^{-1}$) was oscillated with a constant amplitude of 50 pm. The tip apex was functionalized with

a CO-molecule, and all images were captured in constant height mode and a bias sample of 1 mV. AFM images are raw data whereas STM images were subject to standard processes. The images were processed using WSxM[47] software.

**Computational details.** We employed density functional theory (DFT) calculations to analyze atomic and electronic structure for freestanding infinite systems of anthracene and bisanthene polymers presented in Fig. 3 using FHI-AIMS packages[48] using B3LYP[49] exchange-correlation functionals. Systems were allowed to relax until the remaining atomic forces reached below $10^{-2}$ eV Å$^{-1}$. For all infinite systems with periodic boundary condition (PBC), a Monkhorst-Pack grid of 18 x 1 x 1 was used to sample the Brillouin zone. Theoretical d*I*/d*V* maps were calculated by FHI-AIMS with Probe Particle SPM code[50] s-like orbital tip[51]. Vibrational modes of freestanding anthracene and bisanthene polymers featuring conformation IMr1 (Supplementary Fig. S5) and consisting of four acene/periacene units were calculated using dynamical matrix approach with Fireball code[28].

To analyze the optimal reaction pathways, we used a QM/MM Fireball/Amber[29] method based on the combination of classical force-field techniques with Amber[52] and local orbital DFT with Fireball[28]. In the MM part consisting of slab ($40 \times 20$) with three layers, we used the interface force-field[53]. In the QM region consisting of polymer and select gold atoms of surface placed under the bridging unit where the reaction took place (see Supplementary Fig. 2a, b), we used the Fireball DFT method with BLYP exchange-correlation functional[54] with D3 corrections[55] and norm-conserving pseudopotentials. We employed a basis set of optimized numerical atomic-like orbitals (NAOs)[56] with a 1 s orbital for H and sp$_3$ orbitals for C atoms. Note that Fireball DFT-BLYP calculations provide very similar atomic and electronic structure of both anthracene and bisanthene polymers as FHI-AIMS DFT-B3LYP calculations. Before free energy profile calculations, we have performed a QM/MM geometry relaxation followed by a thermalization of the system from 100 to 500 K in order to equilibrate the system. The free energy profile was performed with the WHAM method[57]. In each window of WHAM, we run a QM/MM MD of 2000 steps at 650 K with a time step of 0.5 fs. Calculated NICS(1)$_{zz}$ values shown in Fig. 3c were evaluated using GIAO method[58] at spin-unrestricted ωB97X-D[59]/cc-pVDZ[60] level of theory in Gaussian09[61]. Tetrameric molecules were relaxed at the same level prior to NMR calculations.

## Data availability

The data that support the findings of this study are available from the corresponding authors upon reasonable request.

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

## Acknowledgements

We acknowledge financial support from the Comunidad de Madrid [project QUIMTRONIC-CM (Y2018/NMT-4783)]. This project has received funding from the European Research Council (ERC) under the European Union's Horizon 2020 research and innovation program (grant agreement No 766555). MINECO of Spain (projects CTQ2017-83531-R and RED2018-102815-T) is also acknowledged. IMDEA Nanociencia thanks support from the "Severo Ochoa" Program for Centers of Excellence in R&D (MINECO, Grant SEV-2016-0686). We also acknowledge support from Praemium Academie of the Academy of Science of the Czech Republic, GACR 18-09914S and Operational Program Research, Development and Education financed by European Structural and Investment Funds and the Czech Ministry of Education, Youth and Sports (Project No. CZ.02.1.01/0.0/0.0/16_019/0000754). A.M. and B.M. acknowledge the support from the Internal Student Grant Agency of the Palacký University in Olomouc, Czech Republic IGA_PrF_2020_022 and IGA PrF 2020_034.

## Author contributions

N.M., P.J., and D.E. conceived and designed the experiments. P.J., N.M., and D.E. supervised the project and led the collaboration efforts. B.T., A.S.-G., B.C., B.M., and D.E. carried out the experiments and obtained the data. J.S., E.R.-S., and N.M. synthesized the precursors. The experimental data were analyzed by B.T., A.M., A.S.-G., B.C., K.L., A.B., R.M., P.J., and D.E. and discussed by all the authors. A.M., J.M., S.E., M.O., M.M., and P.J. performed the calculations. The manuscript was written by B.T., N.M., P.J., and D.E. with contribution from all the authors.

## Competing interests

The authors declare no competing interests.
