## [Peer Review File · Nature Communications]

Reviewer #1 (Remarks to the Author):

Synthetic chemistry is based on creating new compounds by rearranging the connectivity of the atoms involved in the reaction. Excitation of vibrational modes and electronic states in the reactant molecules could promote rearrangements that determine the reaction path.

In the present communication the authors report the on-surface synthesis of a pentalene-bridged ladder polymer and characterize the aromaticity and the band gap of the resulting product. To investigate the system the authors use a combination of state-of-the-art scanning probe microscopy featuring sub-molecular resolution and QM/MM calculations. The novelty of the communication lies on the concepts used to interpret the reaction paths of two structurally related precursors (featuring 1 and 2 anthracene units). The analysis of the vibrational modes and the transition states shed light onto the reaction mechanism resulting on the pentalene-bridged ladder polymer.

The main claim of the paper: *the introduction of a new synthetic paradigm that is based on the search of specific vibrational modes by tailoring the π conjugation of the precursors to affect the attempt frequency of a chemical reaction*, lies on the observation that the incorporation of a second anthracene unit in the precursor monomer affects the π conjugation and therefore the vibrational landscape of the precursor bisanthene polymer. Therefore, when heating the bisanthene polymer it reaches a transition state that features vibrational modes overlapping with the cyclization reaction coordinates, favoring the intra-polymeric cyclization pathway.

In my view, the interpretation of the results is consistent with the data. The experimental data is technically sound and can be reproduced with the details given by the authors. The methodology is adequate to support the main claims of the work. In particular, the interpretation based on the vibrational modes of the intermediates matching the reaction coordinates to form the product is likely to influence the thinking in the field of *on-surface synthesis* in particular, and organic chemistry in general. Therefore, I support the publication of the manuscript as a Communication after some minor points are addressed.

My main criticism of the work in the present form is that the role of the supporting surface is totally neglected in the discussion. The authors have certainly reflected on the role of the surface and made a number of sound assumptions, like introducing a gold adatom to abstract the hydrogen. I strongly encourage the authors to openly discuss the limitations of their freestanding computational calculations regarding the role of the Au(111). Au(111) might act as a template to guide/align the intermediates in a particular configuration. Please discuss the assumptions you make as it is expected that the registry of the molecular intermediates on the surface (molecule-surface interaction) might also influence the vibrational modes and the diffusion of the molecule on the surface. Do you have any evidence to neglect the geometrical and the electronic properties of the surface on the mechanism? Please explain.

- In lines 105-106 the authors claim that the ladderization reaction is 100% selective and no-side reactions are detected at sub-monolayer coverage. Such a statement cannot be concluded from the single image of about 20 nm shown on Fig 1d. Please either provide experimental evidence in the supplementary showing large scale images featuring a single product or remove that statement.

- How many modes were calculated for each precursor polymer? Why figure S5 shows only 4 vibrational modes for the anthracene polymer? Please provide more details on Fig S5. Does it consider only vibrational motions of the reactant molecule in its starting equilibrium configuration?
- Concerning the aromaticity of the ladder compound, I assume all the readers will be familiar with the Hückel's rules, some with the Clar's sextets and almost nobody with the Glidewell and Lloyd rules. I suggest to guide the reader by focusing on the concepts first and then the names. For example, on 214-215 it will be helpful to write that the Glidewell and Lloyd rule is an extension of the original Hückel's $4n + 2$ rule that is valid for polycyclic non-benzenoid hydrocarbons, and then state the rule. In 224 briefly define the Clar's sextets. These minor changes will help all the readers to easily follow the discussion.

Minor points:

- Figure S2 is called in the text after Figure S3
- 153- Semi-transparent or translucent color
- Fig. S3, S4 define the Au adatom (yellow ball) in the caption. The schemes show the surface, is the registry of the molecule to the Au(111) based on any calculation?
- Please include in the experimental details how long was the annealing to 650-700K.

Reviewer #2 (Remarks to the Author):

The authors provide a comparison of the reactivity associated with two closely related pi-conjugated polymers. Whilst one of them transforms into pentalene-bridged ladder polymers upon annealing, the other does not. The differences are rationalized in terms of the different vibrational modes exhibited by the two polymers.

The manuscript thus includes many very interesting points. The polymers studied are of great interest themselves. Although the starting polymers have been readily published earlier, that is not the case for the pentalene-bridged ladder polymers. The reaction mechanism to form these polymers is new and interesting as well. However, the correlation of the reactivity with specific vibrational modes along the reaction coordinates is most novel and important, since it may be the key to advance in the understanding of readily demonstrated reactions and use it for further refinements of on-surface synthesis strategies. Altogether, supported by excellent quality data and a scholarly presentation, the paper fulfils the requirements in terms of novelty and importance for publication in Nature Communications.

There are, however, several aspects that should be improved before publication:

- There are multiple parts where the phrasing could be improved. Just focusing on the abstract the parts "...a theoretical and experimental synthetic paradigm...", "...we on-surface design...", "...specific pi-topology..." or "...and free-defect pentalene-bridged..." should be revised. It continues in the introduction with the first sentence, "... great potential application in..." should be changed to "great potential for applications..." or alike.
- The authors speak about the free energy (e.g. in line 134). Is it really free energy or rather enthalpy? If truly free energy, the authors should provide information on how the entropy has been included.

- Fig. 2 requires an energy scale as y axis that makes the differentiation between total energy and energy barriers more obvious. As it is, the meaning of the numbers is initially somewhat confusing.
- The guiding thread throughout the paper is the Arrhenius equation, justifying the different reactivity by changes of its two parameters, the attempt frequency and the activation energy. Although a very intuitive and didactic equation, it is a mere phenomenological model, in which neither the attempt frequency nor the activation energy are associated to any well-defined physical quantity. Since the work provides such a nice correlation of the ab-initio calculated quantities (vibrational modes and transition state energies for the various reaction steps) with the reactivity, it would be very instructive to have it correlated to a rate equation based on other well-defined quantities equally accessible by DFT, as is the closely related Eyring equation. Although I see the advantages (simplicity) for maintaining the Arrhenius equation as the guiding thread, at least some discussion about that would be very valuable for the overall message.
- The authors correlate the different “attempt frequency” factors between the two polymers to a change in the frequency of the vibrational modes and to their overlap with the reaction coordinate (lines 164-170). They should add an educated guess of which of these two factors is dominant, not explicitly stated right now.
- Line 210: the explanation of cis and trans is very confusing. It would be much better to leave it with the reference to the figure. Otherwise, please rewrite.
- The authors jump to the conclusion that Glidewell and Lloyd’s rules apply to this experimental case from the match of experimental and theoretical bond lengths. This seems too early in the manuscript. Bare DFT, unless you analyze and argue why some bonds are shorter or longer (which is not the case in the current manuscript version), does not tell you about the aromaticity. Instead, NICS does (a ref. is missing when introducing NICS, by the way). The sentence of lines 232 and 233 should thus better be moved to a position after presentation of the NICS results.
- It would help to make more clear which are the theory and which the experimental images in Fig. 3f.

Reviewer #3 (Remarks to the Author):

The authors report on the synthesis of a new, ladder type conjugated polymer that incorporates non-benzenoid moieties. In a comparative study of two different precursors, anthracene and bisanthracene derivatives, they observe that the cyclization of the intermediate polymer that leads to the non-benzenoid moieties only occurs in one of them (the one with the bisanthracene derivative). Both of the intermediate polymers having similar cyclization activation barriers, they attribute the differences to the fact that the different π -conjugation in each case lead to different bonding order of the ethynylene bridges. This leads to the presence/absence of particular bending modes that enhance the attempt frequency of the reaction. The main conclusion of the work is that a full picture that combines the activation energy landscape and the vibrational structure can lead to a more predictive on-surface synthesis of organic covalent nanostructures. Also that tailoring vibrational properties of intermediates can activate reaction paths that lead to new structures. The study is an excellent example of how advanced SPM techniques combined with ab-initio calculations can lead to a profound understanding of on-surface chemical reactions that can be crucial for the advance in this field, and the results are very relevant to the hot topic of on-surface synthesis of

carbon nanostructures. The manuscript is very clearly written, and the experimental data is convincing and of very high quality. Thus, I support publication in this journal, after addressing some minor comments:

1.- Introduction: The list of references used to describe the limited number of strategies to synthesize conjugated polymers is not reflecting the literature in a balanced way: the authors use two self-references that contain the same systems studied in this work, and then summarize the rest of the work in a review of graphene nanoribbons that is not complete (it is from 2016, there is a chapter in the book of On-surface Synthesis II edited by Springer in 2018 with a more complete overview on the on-surface synthesis of graphene nanoribbons), and a single additional example of another conjugated polymers. The authors should extend examples of other complex structures. Some examples are other non-benzenoid chains (e.g. Sánchez-Sánchez et al. Chem. – A Eur. J. 25, 12074–12082 (2019)), 2D conjugated polymers (e.g. Moreno et al. Science 360, 199–203 (2018), Steiner et al. Nat. Commun. 8, 14765 (2017)), functionalized/doped polymers (e.g. J. Cai et al. Nat. Nanotechnol. 9, 896–900 (2014), J. Li et al. ACS Nano. 14, 1895–1901 (2020)), etc.

2.- For the activation barrier of the blue reaction path for formation of pentalene with the anthracene derivatives, a value of >100 kJ/mol is given, but I cannot derive this value from Fig. S4, I see two steps of 37 kJ/mol, and 36 kJ/mol respectively.

3.- In the same discussion (pg 10), I do not understand the argumentation used to explain the presence of defects in the anthracene-based structures. The authors propose similar probabilities for the blue (double dehydrogenation) and yellow (dehydrogenation-cyclization-dehydrogenation) paths due to the balance between energy barriers and enabling vibrational mode frequencies, which is very reasonable. But why having two channels for the formation of pentalene should lead to other type of structures? I don't see why this fact should lead to the observed defects.

4.- I don't find the description of the unit cell including substrate used in the calculations. How many Au layers were used?

5.- STM is an ideal tool to study chemical reactions at the single molecule level, being capable of exploring vibrational selective processes by the action spectroscopy. Inducing pentalene formation in the bisanthrene polymers by inducing the specific vibrational modes suggested by theory by inelastic tunnelling electrons (and showing that this cannot be induced in the anthracene case) would be a conclusive proof the main conclusion of this work. Did the authors try to do it? Can they comment on it?

REVIEWER 1.

Synthetic chemistry is based on creating new compounds by rearranging the connectivity of the atoms involved in the reaction. Excitation of vibrational modes and electronic states in the reactant molecules could promote rearrangements that determine the reaction path.

In the present communication the authors report the on-surface synthesis of a pentalene-bridged ladder polymer and characterize the aromaticity and the band gap of the resulting product. To investigate the system the authors use a combination of state-of-the-art scanning probe microscopy featuring sub-molecular resolution and QM/MM calculations. The novelty of the communication lies on the concepts used to interpret the reaction paths of two structurally related precursors (featuring 1 and 2 anthracene units). The analysis of the vibrational modes and the transition states shed light onto the reaction mechanism resulting on the pentalene-bridged ladder polymer.

The main claim of the paper: *the introduction of a new synthetic paradigm that is based on the search of specific vibrational modes by tailoring the π conjugation of the precursors to affect the attempt frequency of a chemical reaction*, lies on the observation that the incorporation of a second anthracene unit in the precursor monomer affects the π conjugation and therefore the vibrational landscape of the precursor bisanthene polymer. Therefore, when heating the bisanthene polymer it reaches a transition state that features vibrational modes overlapping with the cyclization reaction coordinates, favoring the intra-polymeric cyclization pathway.

In my view, the interpretation of the results is consistent with the data. The experimental data is technically sound and can be reproduced with the details given by the authors. The methodology is adequate to support the main claims of the work. In particular, the interpretation based on the vibrational modes of the intermediates matching the reaction coordinates to form the product is likely to influence the thinking in the field of *on-surface synthesis* in particular, and organic chemistry in general. Therefore, I support the publication of the manuscript as a Communication after some minor points are addressed.

We thank the referee very much the time devoted for our manuscript, the very high appreciation of our work recommending publication, and his/her insightful comments, which has been replied point-to-point below, altogether re-enforcing the quality of our manuscript.

COMMENT 1.

My main criticism of the work in the present form is that the role of the supporting surface is totally neglected in the discussion. The authors have certainly reflected on the role of the surface and made a number of sound assumptions, like introducing a gold adatom to abstract the hydrogen. I strongly encourage the authors to openly discuss the limitations of their freestanding computational calculations regarding the role of the Au(111). Au(111) might act as a template to guide/align the intermediates in a particular configuration. Please discuss the assumptions you make as it is expected that the registry of the molecular intermediates on the surface (molecule-surface interaction) might also influence the vibrational modes and the diffusion of the molecule on the surface. Do you have any evidence to neglect the geometrical and the electronic properties of the surface on the mechanism? Please explain.

We thank the referee for his/her comment, which clearly demonstrates that the description of the simulations was not clear in the original version of the manuscript. Calculated electronic

and atomic structure presented in Fig. 3 were obtained for free-standing infinite pentalene polymer. From our previous studies, we learned that the free-standing polymer fits reasonably well the experimental measurements of ethynylene-bridged acene polymers on Au(111) surface (Cirera *et al.* Nature Nano. 15, 437-443 (2020)). This can be justified by relatively weak interaction between polymer and metallic substrate, which is mostly driven by the dispersive interaction.

Nevertheless, all free energy calculations were carried out employing QM/MM method including not only the polymer, but also selected number of surface atoms in a region where the chemical reaction takes place, which were included in the QM part. Namely, in the case of reaction pathways involving gold adatoms, the quantum region contained beside the polymer also the adatoms and selected gold atoms underneath, see new Supplementary Fig. S2. In the present version of the manuscript, we modified description of the computational methods and added Supplementary Fig. S2.

COMMENT 2.

In lines 105-106 the authors claim that the ladderization reaction is 100% selective and no-side reactions are detected at sub-monolayer coverage. Such a statement cannot be concluded from the single image of about 20 nm shown on Fig 1d. Please either provide experimental evidence in the supplementary showing large scale images featuring a single product or remove that statement.

We appreciate the referee's comment. We have reworded such statement. The new phrase reads as follows: "Notably, this intrapolymeric ladderization reaction is highly selective and only minority concomitant defects are detected for submonolayer coverage".

COMMENT 3.

How many modes were calculated for each precursor polymer? Why figure S5 shows only 4 vibrational modes for the anthracene polymer? Please provide more details on Fig S5. Does it consider only vibrational motions of the reactant molecule in its starting equilibrium configuration?

Supplementary Fig. S5 displays only those calculated vibrational modes of the initial IMr1 configuration, which have significant contribution of the bending mode located on the bridging unit. To identify these modes, we calculated all vibrational modes employing the dynamical matrix calculations of the free-standing IMr1 model consisting of 4 anthracene/bisanthene units. In total it was 285 (3*95 atoms) and 507 (3*169 atoms) modes for anthracene and bisanthene model, respectively.

COMMENT 4.

Concerning the aromaticity of the ladder compound, I assume all the readers will be familiar with the Hückel's rules, some with the Clar's sextets and almost nobody with the Glidewell and Lloyd rules. I suggest to guide the reader by focusing on the concepts first and then the names. For example, on 214-215 it will be helpful to write that the Glidewell and Lloyd rule is an extension of the original Hückel's $4n + 2$ rule that is valid for polycyclic non-benzenoid hydrocarbons, and then state the rule. In 224 briefly define the Clar's sextets. These minor changes will help all the readers to easily follow the discussion.

We appreciate very much the comment from the referee. We have amended the main paragraph addressing the Glidewell and Lloyd's rules. In addition, we have marked in blue in Figure 3a the Clar's sextets. The improved paragraph reads as follows: "Clar's π -sextet rule, an extension of Huckel's rule for polycyclic aromatic hydrocarbons (PAHs), states that the Kekulé resonance structure with the largest number of disjoint aromatic π -sextets, i.e., benzene-like moieties, is the most important resonance structure for the characterization of the PAHs properties. Since our pentalene bridged polymers present non-benzenoid units, the rule to be used here is an extension of Clar's π -sextet rule, the one proposed by Glidewell and Lloyd^{37,38}. This rule states that the total population of π -electrons in conjugated polycyclic systems tends to form the smallest $4n+2$ groups and to avoid the formation of the smallest $4n$ groups. The pentalene bridge can show two types of resonant forms, i.e. the inner bond can be single or double. From this assumption, at least two distinct resonant forms can be plotted as illustrated in Figure 3a. By applying the Glidewell and Lloyd's rule, it is evident that the conjugated structure associated with an inner single bond (black structure in Figure 3a) should be the most stable, since the other option (grey structure in Figure 3a) would imply the formation of 4 groups with 4π -electrons in the bisanthene moiety, which must be avoided according to the rule. Interestingly, by following the observed π -conjugation, the system stabilizes four Clar's sextets in the bisanthene (depicted in blue in Figure 3a), the maximum number."

MINOR POINTS.

COMMENT 5. Figure S2 is called in the text after Figure S3.

We thank the referee. Following her/his comment, we revised the numbering of the figures and now they are ordered upon appearance in the text.

COMMENT 6. 153- Semi-transparent or translucent color

We corrected the expression accordingly.

COMMENT 7. Fig. S3, S4 define the Au adatom (yellow ball) in the caption. The schemes show the surface, is the registry of the molecule to the Au(111) based on any calculation?

Yes, as we mentioned above the QM/MM calculations include both anthracene/bisanthene polymer and Au(111) surface. Few atoms underneath of the bridging units where the reaction takes place were included in the QM region. The rest of surfaces atoms was treated on the MM level. Thus, the new Figure S2 displays the geometries employed in the QM/MM calculations.

COMMENT 8. Please include in the experimental details how long was the annealing to 650-700 K.

The duration of the annealing step is now included in the experimental details.

REVIEWER #2

The authors provide a comparison of the reactivity associated with two closely related π -conjugated polymers. Whilst one of them transforms into pentalene-bridged ladder polymers upon annealing, the other does not. The differences are rationalized in terms of the different vibrational modes exhibited by the two polymers.

The manuscript thus includes many very interesting points. The polymers studied are of great interest themselves. Although the starting polymers have been readily published earlier, that is not the case for the pentalene-bridged ladder polymers. The reaction mechanism to form these polymers is new and interesting as well. However, the correlation of the reactivity with specific vibrational modes along the reaction coordinates is most novel and important, since it may be the key to advance in the understanding of readily demonstrated reactions and use it for further refinements of on-surface synthesis strategies. Altogether, supported by excellent quality data and a scholarly presentation, the paper fulfils the requirements in terms of novelty and importance for publication in Nature Communications.

We thank the referee for the time devoted to our manuscript, the highly appreciation of our work and his/her suggestion of publication after point-to-point revision to his/her comments, which has improved the quality of the manuscript.

There are, however, several aspects that should be improved before publication:

COMMENT 1.

- There are multiple parts where the phrasing could be improved. Just focusing on the abstract the parts "...a theoretical and experimental synthetic paradigm...", "...we on-surface design...", "...specific π -topology..." or "...and free-defect pentalene-bridged..." should be revised. It continues in the introduction with the first sentence, "... great potential application in..." should be changed to "great potential for applications..." or alike.

We appreciate the referee's comment and following the suggestions we have amended the criticised sentences in the abstract and the introduction.

The new abstract reads as follows: "The development of synthetic strategies to engineer π -conjugated polymers is of paramount importance in modern chemistry and materials science. Here we introduce a synthetic protocol based on the search for specific vibrational modes through an appropriate tailoring of the π -conjugation of the precursors, in order to increase the attempt frequency of a chemical reaction. First, we design a 1D π -conjugated polymer on Au(111), which is based on bisanthene monomers linked by cumulene bridges that tune specific vibrational modes. In a second step, upon further annealing, such vibrational modes steer the two-fold cyclization reaction between adjacent bisanthene moieties, which gives rise to a long pentalene-bridged conjugated ladder polymer featuring a low band gap. In addition, high resolution atomic force microscopy allows us to identify by atomistic insights the resonance form of the polymer, thus confirming the validity of the Glidewell and Lloyd's rules for aromaticity. This on-surface synthetic strategy may stimulate exploiting previously precluded reactions towards novel π -conjugated polymers with specific structures and properties."

The first sentence of the introduction reads as follows: “The design and synthesis of π -conjugated polymers is a very active area of research with great potential for applications in organic field-effect transistors (OFETs), photovoltaics (OPVs) and light emitting diodes (OLEDs)^{1,2,3}.”

COMMENT 2.

The authors speak about the free energy (e.g. in line 134). Is it really free energy or rather enthalpy? If truly free energy, the authors should provide information on how the entropy has been included.

Indeed, we calculated the free energy profile using a standard approach, so-called weighted histogram analysis method (WHAM) method, for details about the method, see ref. 58 in the manuscript.

COMMENT 3.

Fig. 2 requires an energy scale as y axis that makes the differentiation between total energy and energy barriers more obvious. As it is, the meaning of the numbers is initially somewhat confusing.

We modified the scheme accordingly. Now energy barriers are shown in brackets in Figure 2.

COMMENT 4.

The guiding thread throughout the paper is the Arrhenius equation, justifying the different reactivity by changes of its two parameters, the attempt frequency and the activation energy. Although a very intuitive and didactic equation, it is a mere phenomenological model, in which neither the attempt frequency nor the activation energy are associated to any well-defined physical quantity. Since the work provides such a nice correlation of the ab-initio calculated quantities (vibrational modes and transition state energies for the various reaction steps) with the reactivity, it would be very instructive to have it correlated to a rate equation based on other well-defined quantities equally accessible by DFT, as is the closely related Eyring equation. Although I see the advantages (simplicity) for maintaining the Arrhenius equation as the guiding thread, at least some discussion about that would be very valuable for the overall message.

We thank the referee for this intriguing comment, which touches very fundamental question: “*Is it the transition state theory valid for such systems and/or how vibrational modes of the initial state affect the reaction rate?*”. Indeed, we intentionally referred in the original version of the manuscript to the Arrhenius equation instead of the transition state theory to underline the importance of vibrational modes of the initial state (reactant) on the reaction rate.

Strictly speaking, in the transition state theory (including the Eyring equation) the rate of a reaction is determined by dynamics of the system at the transition state (i.e., dividing surface, which separates the phase space into a reactant and product region, placed at the lowest saddle point of potential energy surface). Importantly, one of the basics assumptions of the transition state theory is the existence of a quasi-equilibrium in reactant region (i.e. between the reactants and the activated complex formed at the transition state). This means that the reactants have sufficient kinetic energy which allows to reach the transition state (i.e. to form

the activation complex) at certain frequency to establish the quasi-equilibrium. In the case of bimolecular reactions in gas phase, this kinetic energy is often available through collision events of reactants. Therefore, such equilibrium can be reached depending on collision rates driven by temperature of system or kinetic energy of a molecular beam. However, in the case of unimolecular reactions caused by an internal chemical transformation of the reactant, such as our case, the kinetic energy is driven mostly by internal vibrational modes of the reactant. Moreover, only vibrational modes with certain overlap with the reaction coordinates may contribute to the kinetic energy required to reach the transition state. From this thought line, it seems that not only the transition state but also the internal vibrational modes of the reactant may play important role in such reaction mechanisms.

Unfortunately, we do not have yet in hands a solid and rigorous theory which would allow us to provide some quantitative numbers. This requires long term effort and it is subject of our current research. However, we feel that the presented combined experimental and theoretical work provides a strong indication that indeed the vibrational modes of the initial state may play significant role on the rate. We have to say, that this idea is not completely new and it can be tracked down to early works of Polanyi introducing a concept of “early” and “late” reaction barriers (see e.g. *J.C. Polanyi Accounts of Chem. Res.* 5, 161 (1972)) or more recently control of reaction dynamics via excitations of reactant vibrational modes (*F.F. Crim PNAS*, 105, 12654-12661 (2008)), modified versions of the transition state theory (e.g. *J.L. Bao, D.G. Truhlar Chem. Soc. Rev.* 46, 7548 (2017)) including mode-specific reaction dynamics (*H. Guo, K. Liu, Chem. Sci.* 7, 3992-4003 (2016)). However, these recent theories have been tested to bimolecular reactions including reactants, much simpler than such polymers presented here. As this, we hope that the presented example will stimulate new direction of both experimental and theoretical activities towards better understanding of the chemical reaction rates, especially those on-surface reactions driven by internal vibrational modes.

To clarify this point in the main text, a new paragraph has been included which read as follows:

“The discussion above points out the importance of the internal modes of the reactant on the chemical rate of a given reaction. From this perspective, it is worth to make a connection to well-established transition state theory³⁵, which is widely adopted to describe the chemical rates of diverse reaction mechanisms. Strictly speaking, in the transition state theory the rate of a reaction is determined by dynamics of the system at the transition state (i.e. dividing surface, which separates the phase space into a reactant and product region, placed at the lowest saddle point of potential energy surface). Importantly, one of the basic assumptions of the transition state theory is the existence of a quasi-equilibrium in the reactant region (i.e. between the reactants and the activated complex formed at the transition state). This assumption means that the reactants have sufficient kinetic energy, which allows to reach the transition state (i.e. to form the activation complex) at certain frequency to establish the quasi-equilibrium. In the case of bimolecular reactions in gas phase, the required kinetic energy is often available through collision events of reactants. Therefore, such equilibrium can be reached depending on collision rates driven by temperature of system or kinetic energy of a molecular beam. However, in the case of unimolecular reactions caused by an internal chemical transformation of the reactant, such as the one discussed above, the kinetic energy is driven mostly by internal vibrational modes of the reactant. Moreover, only vibrational modes with certain overlap with the reaction coordinates may contribute to the kinetic energy required to reach the transition state. Thus, it seems that not only the transition state, but also

the internal vibrational modes of the reactant may play important role in such reaction mechanisms. We feel that our combined experimental and theoretical work provides a strong indication that, indeed, the vibrational modes of the initial state may play significant role on the rate. It is worth to point out that our rationalization roots back to early works from Polanyi introducing a concept of “early” and “late” reaction barriers³⁴, or, more recently, to control of reaction dynamics via excitations of reactant vibrational modes³², and to modified versions of the transition state theory³⁶ including mode-specific reaction dynamics³³. However, these recent theories have been tested to bimolecular reactions including reactants, much simpler than the polymers presented here.”

COMMENT 5.

The authors correlate the different “attempt frequency” factors between the two polymers to a change in the frequency of the vibrational modes and to their overlap with the reaction coordinate (lines 164-170). They should add an educated guess of which of these two factors is dominant, not explicitly stated right now.

In this particular case, the difference between frequencies of the vibrational modes of anthracene and bisanthene steering the transformation towards the pentalene bridged polymer is relatively small (~ 2MHz, i.e. 10% difference). On the other hand, the analysis of the projection of the vibrational modes on the reaction coordinates reveals in the case of the anthracene polymer the lack on the in-phase motion of unsaturated carbon atoms with the bending mode of the ethynylene linker reducing substantially the projection. Therefore, in this particular case, the overlap between the vibrational mode and the reaction coordinate plays the dominant role.

On the other hand, the frequency factor may play a leading role when comparing chemical reactions with similar activation energies but driven by different vibrational modes with quite distinct frequencies, such as comparison of hydrogenation abstraction (driven by C-H stretching mode 3100 cm^{-1} i.e. 92 MHz) and the cyclization reaction (driven by C-H stretching mode $\sim 600\text{ cm}^{-1}$ i.e. 18 MHz) of the anthracene polymer in the manuscript.

We accordingly extended the discussion in the main text with the addition of the following paragraph:

“Therefore, in the case of the cumulene-like bridged bisanthene polymer (2) the strong overlap of the bending vibrational modes with the cyclization reaction coordinates together with slightly higher vibrational frequencies provide larger attempt frequency ν than in the case of ethynylene-linked anthracene polymer (1). Indeed, the bending modes of ethynylene-bridge of the anthracene polymer (1) are completely lack of the in-phase motion of the unsaturated carbon atoms, see Supplementary Fig. S5, which plays the important role in the formation of the pentalene bridge. As a result, the overlap between the cyclization reaction coordinate and the internal vibrational modes of the polymer is significantly reduced, approx. $\sim 30\%$ with respect to the bisanthene case. Consequently, this causes substantial drop of the corresponding attempt frequency ν . Furthermore, the anthracene polymer (1) contains the ethynylene bridges, thus featuring a triple bond, which has distinct bonding character than the cumulene-like linkers of bisanthene polymer (2)^{27,28}. Therefore, the frequencies of the steering bending modes of the ethynylene linkers in anthracene polymers (1) are lower by 50-100 cm^{-1} (i.e. a drop of $\sim 10\%$) than those of cumulene-like bridges in bisanthene polymers

(2), as shown in Supplementary Fig. S5, leading to a further decrease of the attempt frequency.”

COMMENT 6.

Line 210: the explanation of cis and trans is very confusing. It would be much better to leave it with the reference to the figure. Otherwise, please rewrite.

We thank the referee for his/her comment. We followed the suggestion and leave the sentence with the reference to the figure.

COMMENT 7.

The authors jump to the conclusion that Glidewell and Lloyd's rules apply to this experimental case from the match of experimental and theoretical bond lengths. This seems too early in the manuscript. Bare DFT, unless you analyze and argue why some bonds are shorter or longer (which is not the case in the current manuscript version), does not tell you about the aromaticity. Instead, NICS does (a ref. is missing when introducing NICS, by the way). The sentence of lines 232 and 233 should thus better be moved to a position after presentation of the NICS results.

We appreciate the reviewer's comment. The logic we followed in those paragraphs was to introduce the Glidewell and Lloyd's rules for non-benzenoid polycyclic hydrocarbons, to guide the reader to the most dominant resonance form expected in our case. Then by DFT and nc-AFM we deduce that resonance form. Finally with NICS, nicely pointed out by the referee, we analyze for our specific case how the calculated aromaticity should be. Thus we feel this argumentation is correct and we would prefer to keep the ordering as it is.

COMMENT 8.

It would help to make more clear which are the theory and which the experimental images in Fig. 3f.

We thank the referee's comment. We have modified the figure accordingly.

REVIEWER #3

The authors report on the synthesis of a new, ladder type conjugated polymer that incorporates non-benzenoid moieties. In a comparative study of two different precursors, anthracene and bisanthracene derivatives, they observe that the cyclization of the intermediate polymer that leads to the non-benzenoid moieties only occurs in one of them (the one with the bisanthracene derivative). Both of the intermediate polymers having similar cyclization activation barriers, they attribute the differences to the fact that the different π -conjugation in each case lead to different bonding order of the ethynylene bridges. This leads to the presence/absence of particular bending modes that enhance the attempt frequency of the reaction. The main conclusion of the work is that a full picture that combines the activation energy landscape and the vibrational structure can lead to a more predictive on-surface synthesis of organic covalent nanostructures. Also that tailoring vibrational properties of intermediates can activate reaction paths that lead to new structures. The study is an excellent example of how advanced SPM techniques combined with ab-initio calculations can lead to a profound understanding of on-surface chemical reactions that can be crucial for the advance in this field, and the results are very relevant to the hot topic of on-surface synthesis of carbon nanostructures. The manuscript is very clearly written, and the experimental data is convincing and of very high quality. Thus, I support publication in this journal, after addressing some minor comments.

We thank the referee for the time devoted to our manuscript and the very high appreciation of our work, recommending publication after minor revision. We have addressed point-to-point his/her comments below, which have contributed to the improved of the manuscript.

COMMENT 1.

Introduction: The list of references used to describe the limited number of strategies to synthesize conjugated polymers is not reflecting the literature in a balanced way: the authors use two self-references that contain the same systems studied in this work, and then summarize the rest of the work in a review of graphene nanoribbons that is not complete (it is from 2016, there is a chapter in the book of On-surface Synthesis II edited by Springer in 2018 with a more complete overview on the on-surface synthesis of graphene nanoribbons), and a single additional example of another conjugated polymers. The authors should extend examples of other complex structures. Some examples are other non-benzenoid chains (e.g. Sánchez-Sánchez et al. *Chem. – A Eur. J.* 25, 12074–12082 (2019)), 2D conjugated polymers (e.g. Moreno et al. *Science* 360, 199–203 (2018), Steiner et al. *Nat. Commun.* 8, 14765 (2017)), functionalized/doped polymers (e.g. J. Cai et al. *Nat. Nanotechnol.* 9, 896–900 (2014), J. Li et al. *ACS Nano.* 14, 1895–1901 (2020)), etc.

We appreciate the reviewer's comment. In the paragraph just before the sentence mentioned by the reviewer we included five reviews about on-surface synthesis. To emphasize the literature more explicitly we have expanded the list of references regarding conjugated polymers, including in that sentence the mentioned reviews and the suggested references by the reviewer.

COMMENT 2.

For the activation barrier of the blue reaction path for formation of pentalene with the anthracene derivatives, a value of >100 kJ/mol is given, but I cannot derive this value from Fig. S4, I see two steps of 37 kJ/mol, and 36 kJ/mol respectively.

We apologise for the confusion caused and would like to clarify this point as follows. In Supplementary Figure S4, we show 3 possible reaction pathways of the chemical transformation of anthracene polymer obtained from QM/MM calculations. Indeed, the blue trajectory consisting of two subsequent dehydrogenation processes on adjacent anthracene units leading to IMr2 state with two activation barriers 37 and 36 kJ/mol. However, this pathway leads to a product other than the pentalene bridged polymer, see sketch of defect in Supplementary Fig. S4. Moreover, the QM/MM calculations indicate that the transition state TS6 adopts a completely different atomic structure than that of the bisanthene polymer (see Supplementary Fig. 3), and that the activation barrier of the cyclization mechanism towards the pentalene bridge unit is very large (> 100 kcal/mol, not shown in Supplementary Figure S4). This finding clearly demonstrates that the cyclization reaction coordinates forming the pentalene bridge (see Supplementary Fig. S2) are no longer the optimal ones and other more favourable reaction mechanisms prevail. Consequently, alternative reaction pathways may take place to form other kind of defective structures as found in the experiment.

To understand this behavior, we need to look at the way how the pentalene bridge is formed. Each unsaturated carbon atom on the anthracene unit has to couple to distant carbon atom in the ethynylene bridge. This process requires an in-phase motion of the bending vibrational mode with the distant unsaturated carbon atoms of the anthracene unit, see Fig R1a. Instead, according to our calculations, the IMr2 anthracene state prefers a different reaction mechanism, shown on Fig R1b. Calculated atomic structure of the transition state TS4 shown on Supplementary Fig. S4, reveals that unsaturated carbon atoms on the anthracene unit connects to the nearest carbon atoms on the ethynylene bridge, which leads to different atomic arrangement than pentalene polymer. This scenario fits well to the experimental picture showing formation of defective anthracene chains instead of the pentalene bridged polymers.

Fig. R1 Schematic view of two possible reaction pathways of cyclization reaction for anthracene polymer leading to different transition states a) towards formation of the pentalene bridge b) formation of other type of bridge structure, see TS4 in Supplementary Fig. S4.

COMMENT 3.

In the same discussion (pg 10), I do not understand the argumentation used to explain the presence of defects in the anthracene-based structures. The authors propose similar probabilities for the blue (double dehydrogenation) and yellow (dehydrogenation-cyclization-dehydrogenation) paths due to the balance between energy barriers and enabling vibrational mode frequencies, which is very reasonable. But why having two channels for the formation of pentalene should lead to other type of structures? I don't see why this fat should lead to the observed defects.

We appreciate this important remark. Indeed, it is interesting that two relatively similar anthracene-based structure provide completely different products, as discussed in the previous reply to the comment 2. We think that this difference is caused by the character of the vibrational bending modes which do overlap a little with the reaction coordinate forming the pentalene bridge. Consequently, in the case of two unsaturated carbon atoms, the system prefers the different reaction pathway connecting the unsaturated carbon atoms with the adjacent instead of the distant atoms in the ethynylene bridge, see Fig R1. We think that this difference is caused by enhanced reactivity of two instead of one unsaturated carbon atom and the poor overlap of the bending modes with the reaction coordinates forming the pentalene bridge.

Thus, we accordingly extended the discussion in the manuscript with the addition of the following text to highlight the underlying concept:

“Notably, for this reaction trajectory of the anthracene polymer (1), the formation of the pentalene bridge occurs through quite distinct transition state (compare TS4 in Supplementary Fig. S4 with TS4 in Supplementary Fig. S3 for the bisanthene polymer) featuring very large activation energy (>100 kcal/mol), not shown in Supplementary Fig. S4. The QM/MM simulations indicate that unsaturated carbon atoms in the anthracene unit connect to the nearest carbon atoms on the ethynylene bridge, which leads to a different atomic arrangement than in the bisanthene polymer. Such a process can be understood by the absence of the in-phase motion of the bending vibrational mode with the distant unsaturated carbon atoms of the anthracene unit as discussed previously. This result clearly demonstrates that the cyclization reaction coordinates forming the pentalene bridge from the ethynylene-bridged anthracene polymer (see Supplementary Fig. S2d) are no longer the optimal ones and other more favourable reaction mechanisms are presented.”

COMMENT 4.

I don't find the description of the unit cell including substrate used in the calculations. How many Au layers were used?

We employed 20x40 slab consisting of 3 layers. Note that we performed QM/MM simulations where only few surface atoms were included in the QM region and the rest was in MM region. We include the details in the new Supplementary Fig. S2.

COMMENT 5.

STM is an ideal tool to study chemical reactions at the single molecule level, being capable of exploring vibrational selective processes by the action spectroscopy. Inducing pentalene formation in the bisanthene polymers by inducing the specific vibrational modes suggested by theory by inelastic tunnelling electrons (and showing that this cannot be induced in the anthracene case) would be a conclusive proof the main conclusion of this work. Did the authors try to do it? Can they comment on it?

We thank the referee's comment. We have performed spectroscopy on both polymers, without successful results regarding transformation into pentalene bridges. The reason is probably related to the inherent mechanism of the complex reaction, which cannot be activated just by a local injection of inelastic electrons, since spectroscopy is performed at 4K.

REVIEWERS' COMMENTS:

Reviewer #1 (Remarks to the Author):

I appreciate the work of the authors incorporating the feedback from the referees. In my opinion, the MS has significantly improved in terms of presentation and clarity. I am happy to support publication in the present form.

Reviewer #2 (Remarks to the Author):

The authors have satisfactorily addressed my previous comments. I recommend publication of this revised and improved version in Nature Communications as is.

Reviewer #3 (Remarks to the Author):

The authors have answered all comments satisfactorily. I consider the manuscript ready for publication.

Aitor Mugarza

Prof. David Écija
c/Faraday 9, 28049, Madrid
+34-91-2998855
david.ecija@imdea.org

REVIEWERS' COMMENTS:

Reviewer #1 (Remarks to the Author):

I appreciate the work of the authors incorporating the feedback from the referees. In my opinion, the MS has significantly improved in terms of presentation and clarity. I am happy to support publication in the present form.

We thank very much the referee for his/her work, the high appreciation of our study, recommending publication as it is.

Reviewer #2 (Remarks to the Author):

The authors have satisfactorily addressed my previous comments. I recommend publication of this revised and improved version in Nature Communications as is.

We appreciate very much the time devoted to our work and the acceptance of our study in the present form.

Reviewer #3 (Remarks to the Author):

The authors have answered all comments satisfactorily. I consider the manuscript ready for publication. Aitor Mugarza

We thank Prof. Mugarza the time devoted to our manuscript and we are delighted with his suggestion of publication as it is.

Sincerely yours,

David Écija on behalf of all the authors.